# Characterization of EIAV *env* Quasispecies during Long-Term Passage In Vitro: Gradual Loss of Pathogenicity

**DOI:** 10.3390/v11040380

**Published:** 2019-04-24

**Authors:** Cong Liu, Xue-Feng Wang, Yan Wang, Jie Chen, Zhaohua Zhong, Yuezhi Lin, Xiaojun Wang

**Affiliations:** 1State Key Laboratory of Veterinary Biotechnology, Harbin Veterinary Research Institute of Chinese Academy of Agricultural Sciences, Harbin 150069, China; liucong@caas.cn (C.L.); wangxuefeng@caas.cn (X.-F.W.); 17545106209@163.com (Y.W.); 2Key Laboratory of Special Animal Epidemic Disease of Ministry of Agriculture, Institute of Special Animal and Plant Sciences, Chinese Academy of Agricultural Sciences, Changchun 130112, China; chenjie@caas.cn; 3Department of Microbiology, Harbin Medical University, Harbin 150069, China

**Keywords:** EIAV, SGA (single genome amplification), quasispecies, virulence

## Abstract

As the only widely used live lentiviral vaccine, the equine infectious anima virus (EIAV) attenuated vaccine was developed by in vitro passaging of a virulent strain for 121 generations. In our previous study, we observed that the attenuated vaccine was gradually selected under increased environmental pressure at the population level (termed a quasispecies). To further elucidate the potential correlation between viral quasispecies evolution and pathogenesis, a systematic study was performed by sequencing *env* using several methods. Some key mutations were identified within Env, and we observed that increased percentages of these mutations were accompanied by an increased passage number and attenuated virulence. Phylogenetic analysis revealed that *env* mutations related to the loss of virulence might have occurred evolutionarily. Among these mutations, deletion of amino acid 236 in the V4 region of Env resulted in the loss of one N-glycosylation site that was crucial for virulence. Notably, the 236-deleted sequence represented a “vaccine-specific” mutation that was also found in wild EIAV_LN40_ strains based on single genome amplification (SGA) analysis. Therefore, our results suggest that the EIAV attenuated vaccine may originate from a branch of quasispecies of EIAV_LN40_. Generally, the presented results may increase our understanding of the attenuation mechanism of the EIAV vaccine and provide more information about the evolution of other lentiviruses.

## 1. Introduction

As the simplest member of the lentivirus family, equine infectious anemia virus (EIAV) shares similar features with other lentiviruses, including its genomic structure, life cycle, cell tropism and antigen evolution [1,2]. The uniqueness of EIAV lies in the fact that most infected horses become asymptomatic carriers after recurrent febrile episodes, which are associated with a high viral load and anemia, and thus the horses achieve natural immunologic control [3]. Notably, an attenuated vaccine strain has been developed that successfully controls EIAV prevalence in China. The vaccine was generated by serial passage of a virulent EIAV strain (EIAV_LN40_) for 16 generations in vivo, followed by passaging for as many as 121 passages in vitro [4,5]. Due to these peculiarities, the EIAV attenuation system may provide an ideal model for elucidating the correlates of genomic evolution and immunologic control.

For RNA viruses, mutation swarms are generated rapidly through error-prone viral replication due to inaccuracy of the RNA polymerase [6]. This mutant swarm, which is also termed a viral quasispecies, appears as a whole population with a mutation-selection balance and is significantly involved in viral pathogenesis [7,8]. Viral quasispecies are also essential for viral survival because they yield beneficial viral phenotypes under in vivo selection. Interestingly, the mutation spectra are higher after long-term passage in vitro, which causes a change in virulence [9,10,11]. Hence, the relationship between RNA viral quasispecies evolution and viral pathogenesis is an intriguing problem.

As discussed previously, multi-position mutations located in the LTR and in *env* appeared during the EIAV attenuation process in vitro, especially in the viral *env* gene, which developed eight hyper variable regions located in the V3 and V4 regions [1,12,13,14]. Further analysis showed that these variations were related to viral pathogenesis during the EIAV attenuation process. A virulence-correlated parallel in *env* variation was also observed during the EIAV attenuation process [4]. In another study, we observed that the LTR showed a similar pattern at the population level [15]. However, the LTR is a noncoding region which could not reflect the predominant antigen gene of the EIAV evolutionary pattern. Hence, *env* was chosen as the target of evolutionary selection in this study to address this important question. The aim of this study was to characterize *env* quasispecies evolution and further investigate the related mechanism between virulence attenuation and *env* quasispecies. Our results will be of great interest for understanding the evolutionary mechanism of the EIAV attenuated vaccine and defining vaccine development strategies for other lentiviruses.

## 2. Materials and Methods

### 2.1. Study Subjects

All samples were stored at the Harbin Veterinary Research Institute, Chinese Academy of Agricultural Sciences (CAAS). These samples included a virulent strain (EIAV_LN40_) that caused an infection in horses with 100% mortality at a dose of 1 × 10^5^ TCID50; this strain was initially isolated from an EIA positive horse and passed for 16 generations in horses, resulting in three representative strains (EIAV_DLV34_, EIAV_DLV62_, and EIAV_DLV92_) with 100%, 100%, and 9.1% mortality that stemmed from EIAV passage in vitro for 34, 62, and 92 generations, respectively (Appendix A) [16]. Additionally, we included a vaccine strain (EIAV_DLV121_) that provided 85% protection against EIAV_LN40_ challenge and a full-length infectious molecular clone (pLGFD3-8) constructed from a vaccine strain and then passaged in fetal donkey dermal (FDD) cells for 3 generations. Animal experiments showed that this infectious clone was avirulent [17].

### 2.2. Viral RNA Extraction and cDNA Synthesis

Viral RNA was extracted from 140 μL of the virus samples using the Viral RNA Mini Kit (QIAamp, Dusseldorf, Germany) according to the manufacturer’s protocol. Reverse transcription of RNA to single-stranded cDNA was performed using the SuperScript^TM^ IV Reverse Transcriptase System (Life Technologies, Carlsbad, USA). First, 1 μg of RNA, dNTPs (0.5 mM each), and 0.5 μM of the *env* NR primer (5’-CAGCTACAATGGCAGCTATTATAGCAG-3’; nucleotides (nt) 6702 to 6676 of the EIAV sequence) were incubated for 5 min at 65 °C to denature the RNA secondary structure. First-strand cDNA synthesis was carried out in 20 μL reaction mixtures with 5× SSIV buffer, 5 mM DTT, 2 U/μL of an RNase inhibitor (RNaseOUT) (Life Technologies, Carlsbad, USA), and 10 U/μL of SuperScript^TM^ IV (Life Technologies, Carlsbad, USA). The reaction mixture was incubated at 55°C for 15 min and heated to 80°C for 10 min to complete the reverse transcription reaction, followed by RNase H (Life Technologies, Carlsbad, USA) digestion at 37 °C for 20 min. The synthesized cDNA was used immediately for PCR or stored at −80 °C.

### 2.3. Bulk PCR

The full-length hypervariable region of the *env* cassette was amplified by nested PCR from the viral cDNA. The specific method was as follows. First, 0.6 μL of bulk cDNA was used for the first-round PCR in a 20 μL volume. The PCR was performed using the KOD FX (Toyobo, Osaka, Japan) system, which included 2× PCR buffer, 0.4 mM dNTPs, and 0.1 μM of the *env* NF (5’-GAAGGCCATCAGGGAGGGAAG-3’; (nt) 5240 to 5260) and *env* NR primers. The cycling conditions were as follows: 94 °C for 2 min, followed by 35 cycles of 94 °C for 30 s, 58 °C for 30 s, and 68 °C for 1 min 30 s, and a final extension of 68 °C for 2 min. The second-round PCR was performed using 0.6 μL of the first-round PCR product and the V3-1 (5’-ACAAACATATACAGGACATCT-3’; (nt) 5808 to 5828) and V4-1 (5’-CTCCAATATTCCAAGAAATAC-3’; (nt) 6218 to 6198) primers under the same conditions used for the first-round PCR. The final PCR products were analyzed by 1% agarose gel electrophoresis. The recycled products were ligated into pMD18-T (TaKaRa, Dalian, China). The constructed plasmids were transformed into E. coli DH5α (TaKaRa, Dalian, China) and spread onto a plate. A total of 98 positive clones identified by PCR were sequenced by COME (Jilin, China).

### 2.4. Probe Design and Real-Time PCR

Two specific probes and primers were synthesized by General Biosystems (Anhui, China). The first probe was the 6-carboxy-fluorescein (FAM) probe (probe-FAM: 5’-TGCAGCAAAGTAAAAACACTTGGAT-3’; (nt) 6010 to 6034), which targeted the *env*-Δ236D-phenotype sequence, and the second was the hexachloride fluorescein (HEX) probe (probe-HEX: 5’-TGCAGCAAAGCGATAATAACACTTG-3’; (nt) 6010 to 6034), which targeted the *env*-236D-phenotype sequence. The *env*-Δ236D-phenotype and *env*-236D-phenotype sequences were named depending on the presence of the 236D deletion in the EIAV Env. The DNA fragments of *env*-Δ236D-phenotype and *env*-236D-phenotype were amplified with primer F (5’-ATAGGAGGTAGACTAAATGGTTCAGG-3’; (nt) 5634-5668) and primer R (5’-AAACAAAAAGAATGGAGGTTGGACA-3’; (nt) 6143-6119) and ligated into the pMD-18T (TaKaRa, Dalian, China) vector. Then, the two plasmids were serially diluted to 10^2^ to 10^9^ copies/μL as standards to construct real-time PCR standard curves. The experiments were performed using 1 μM of each primer, 10 μL of premix Ex Taq (TaKaRa, Dalian, China), 0.25 μM of the HEX-labeled or FAM-labeled probe, 2 μL of the cDNA samples and RNase-free water to a final volume of 20 μL. The samples were amplified using TaqMan-based real-time PCR on the ABI Prism 7500 sequence detection system (Applied Biosystems GmbH, Thermo Fisher Scientific, Wilmington, USA) with one cycle at 95 °C (5 min), followed by 40 cycles at 95 °C (15 s), 58 °C (30 s), and 72 °C (20 s). The calculated efficiencies for all primers were determined by dilution experiments and ranged from 97% to 98%; thus, the target sequences were amplified with similar efficiencies. All samples were run with at least three duplicates.

### 2.5. SGA (Single Genome Amplification)

According to the Poisson distribution [18], the DNA dilution yielding positive PCR products in no more than 30% of the wells will contain one amplicon resulting from single molecule amplification in over 80% of these wells. The viral cDNA was serially diluted to yield a single copy by nested PCR as described for the bulk PCR. All products derived from cDNA dilutions conforming to the above description were sequenced with the V3-1 and V4-1 primers by GENEWIZ (Suzhou, China).

### 2.6. Data Processing and Analysis

The sequences generated in this study have been submitted to GenBank (GenBank acc. no. MK268242 to MK268339 for bulk PCR and MK278920 to MK279296 for SGA). A preliminary sequence analysis was conducted using the SeqMan and MegAlign programs from the Lasergene DNA & Protein analysis software (version 7.0, DNASTAR Inc., Madison, WI, USA), the BioEdit Sequence Alignment Editor (version 7.2.5), and ClustalX (version 2.1). A maximum likelihood tree was constructed using MEGA 7.0 [19]. The amino acid alignment was automatically generated by MargFreq (version 1.02).

## 3. Results

### 3.1. General Characteristics of EIAV env Variability during the Course of EIAV Attenuation Using Bulk PCR

To verify the characteristics of EIAV *env* variation, some *env* genes derived from EIAV strains during the critical stages of the attenuation process were analyzed using bulk PCR [20,21]. A phylogenetic tree was constructed based on 98 *env* sequences amplified from EIAV strains with differing virulence (EIAV_LN40_, EIAV_DLV34_, EIAV_DLV62_, EIAV_DLV92_, and EIAV_DLV121_) (the morbidity of all strains showed in Appendix A). The general features of *env* variability are shown in Figure 1. Obviously, individual clusters accompanied by a decrease in virulence were generated. In other words, the evolutionary pattern of the *env* gene was consistent with the loss of virulence. Notably, the sequences derived from EIAV_DLV62_ displayed a separated distribution between EIAV_LN40_ and EIAV_DLV121_. This result suggested that EIAV_DLV62_ was an important viral evolutionary state that was associated with EIAV pathogenicity. The details of the sequences that also displayed several mutations in each cluster are presented in Table 1. The isolates from each generation displayed the same mutation types (insertions and deletions) but had different mutation frequencies and sites. We observed that the proportion of strains carrying this mutation gradually increased with the serial passage number in vitro (EIAV_DLV34_, EIAV_DLV62_, EIAV_DLV92_, and EIAV_DLV121_ = 6.25%, 18.5%, 42.1%, and 100%, respectively). We also noted that most mutations were random except for a successful amino acid deletion (236D) located in the V4 domain of Env compared to the reference strain sequence (EIAV_LN40_), as shown in Figure 2. This mutation was also stable in horses inoculated with EIAV_DLV121_ [22]. Thus, we speculated that this site might represent a vaccine-specific mutation.

### 3.2. Precise Characteristics of the EIAV env Mutation Distribution during the Attenuation Process

Once the vaccine-specific mutation was verified, we wanted to investigate whether this mutation could be detected during serial passage using a higher sensitivity method. These data will further benefit our understanding of the relationship between *env* quasispecies and virulence. Firstly, we adopted double-probe real-time PCR to accurately determine the proportion of strains with the *env*-Δ236D-phenotype and *env*-236D-phenotype sequences (which refer to virus sequences without and with the 236D mutation) in different EIAV generations [23,24]. An *env*-236D-phenotype-specific probe (HEX) and an *env*-Δ236D-phenotype-specific probe (FAM) were designed based on the V4 domain sequences (Appendix A). To avoid the influence of the initial viral copy number on the detection sensitivity, samples containing different initial viral copy numbers were analyzed in parallel (10^2^ copies/μL, 10^3^ copies/μL and 10^4^ copies/μL). As presented in Appendix A, the *env*-Δ236D-phenotype sequence was detected with low copy numbers from all virulent strains (EIAV_DLV34_ and EIAV_DLV62_) except for EIAV_LN40_, and the proportions of strains with the *env*-Δ236D-phenotype gradually increased with the passage number (from 2.47% to 67.5% at the 10^3^ copies/μL concentration), whereas the proportion of strains with the *env*-236D-phenotype decreased concomitantly.

Although a gradual increase in the percentage of vaccine-specific sequences was observed using double-probe real-time PCR, the distribution of this mutation in EIAV_LN40_ was still unclear. The SGA method can avoid the recombination events induced by the polymerase and better detect minor nucleotide changes at the same position; thus, this technique has been performed recently for viral quasispecies analysis [25,26,27]. Thus, we were interested in using SGA to resequence *env* to obtain more information about *env* quasispecies during long-term passage. Samples obtained separately from EIAV_LN40_ (*n* = 88), EIAV_DLV34_ (*n* = 98), EIAV_DLV121_ (*n* = 95) and infectious clone pLGFD3-8 (*n* = 100) were amplified by SGA and aligned. As expected, the 236D mutation was found in the EIAV strains derived from each generation. The proportion of sequences with the 236D mutation best matched the results derived from the bulk PCR (17.0%, 30.6%, 77.9%, and 100%, respectively) (Table 2). Importantly, we detected 236D (15/72) in the EIAV_LN40_ reservoir, which was not observed using bulk PCR. Additionally, the SGA method exhibited a relatively higher mutation detection rate than bulk PCR. Hence, more variants located at the same position in the EIAV population were observed; for instance, multiple substitutions (E, V, K or Q) instead of a single substitution (E) were detected at position 229 in EIAV_LN40_ using the SGA method. In particular, in addition to position 236, we found that three other positions, which were separately located at S235R, N237K, and N246K, presented gradual and successful mutation accumulation with the increasing passage number, as presented in Table 2. Further analysis showed that the deletion of amino acid 237D resulted in the loss of a N-glycosylation site (237NNTW240) and that the N237K/N246K mutations would also be involved in N-glycosylation site formation (Figure 2A,B). We also noted that these key mutations (235, 236, 237, and 247) were absent or present in combination; for instance, the frequency of the combined mutations at sites 236 and 237 was 85.8% (109/127). The proportions of mutation that concurred at 235, 236, and 237 was 57.5% (73/127). Meanwhile, the sequences with mutations at 235, 236, 237, and 246 showed a lower frequency than the other sequences, as shown in Figure 2C. Moreover, the proportion of these combined mutations at sites 235–236, 235–236–237 and 235–236–237–246 gradually increased as the virulence of the strains decreased, especially in 235–236 (shown in Figure 2D). Hence, we speculated the existence of a link between the sites 236 and 237. According to these results, we speculated that long-term passage in vitro not only had a significant effect on the *env* quasispecies compositions but also might contribute to viral attenuation.

### 3.3. The Relationship between EIAV Attenuation In Vitro and env Gene Quasispecies

To better analyze the relationship between virulence and *env* quasispecies, we constructed a maximum likelihood tree using the MEGA 7.0 program based on the SGA results. As presented in Figure 3, two obvious major groups of *env* sequences have evolved, as represented by the *env*-236D-phenotype (indicated with solid circles) and *env*-Δ236D-phenotype (indicated with hollow circles). Then, serial passaging events continued to lead to accumulation of the *env*-Δ236D-phenotype in the whole population, which made the variant distribution even more complex (i.e., the variant distribution of EIAV_DLV34_). Over the course of 121 generations in vitro, the cluster distribution of related sequences taken from EIAV_DLV121_ showed a pattern similar to that of EIAV_LN40_ except for reversal of the predominant phenotype from *env*-236D to *env*-Δ236D at the individual population level. We observed that two typical *env* phenotypes displayed a gradual transition pattern. These results demonstrated that a predominant phenotype transition occurred in the natural virus population under selection, which was consistent with the LTR investigation results [15].

## 4. Discussion

In contrast to the use of artificial mutagenesis to induce RNA viral attenuation, the EIAV live-attenuated vaccine was developed based on natural evolution through consecutive passaging in vitro. There is growing evidence that analyzing virulence determinants from an evolutionary perspective rather than nucleotide mutations in specific virulence genes is very important, particularly for viruses undergoing consecutive passages in vitro [28,29]. Although a quasispecies is well accepted as a key factor for viral virulence [30], the relationship between these factors remains an open question. In other words, not all viruses will lose their virulence even through serial passage in vitro, such as the CDV Rockborn strain generated in canine macrophage cells [31].

In our previous study, we observed that the EIAV attenuated vaccine also existed as a complex quasispecies [15]. Furthermore, our results suggested that the attenuated vaccine might be a result driven by selection of the dominant EIAV variant at the whole population level. However, whether a similar evolutionary pattern existed in the predominant virulence gene of EIAV was not clear. Here, we used the EIAV attenuation system to investigate the relationship between virulence and EIAV *env* gene variation. In the present study, we used three different methods to answer this question. Regardless of the method used, some stable mutations located in the V3 and V4 regions of Env were verified. Compared with those of the other methods, SGA showed the highest detection sensitivity for minor nucleotide changes; for instance, the mutations located at positions 235, 236, 237, 246, 248, and 250 could be verified only using the SGA method. In other words, more variants were detected at the same positions. Several key mutations, including S235R, N237K, and N246K, attracted our attention. Not only did the percentages of these mutations parallel with the increased passage number but these mutations (237NNTW240 and N237K/N246K) could also result in loss of an N-glycosylation site or formation of an adjacent N-glycosylation site. We confirmed that this position in combination with three other N-glycosylation site mutations in Env resulted in reversion of virulence and influenced the neutralizing activity [32]. Further analysis showed that inter-linkages existed among these key mutations. Based on these results, we speculated that the observed pattern of Env variation was consistent with the loss of viral virulence.

In particular, we noted that the *env*-Δ236D-phenotype, which could present as a vaccine-specific mutation, was present in its parental virus reservoir (EIAV_LN40_) at a lower level. Hence, our observations indicate that the EIAV vaccine strain (*env*-Δ236D-phenotype) may preexist in the EIAV population as a minority population; indeed, another study from our laboratory detected vaccine-specific LTR sequences in horses infected with virulent strains [15]. According to these results, we proposed a model describing *env* evolution during the attenuation process, as shown in Figure 4. In this model, the *env*-236D-phenotype is the predominant hypervirulent EIAV quasispecies that causes typical EIA symptoms in infected horses and induces 100% mortality. The *env*-Δ236D-phenotype is an avirulent quasispecies that may be a beneficial phenotype for new culture environments (equine macrophages), but the underlying reason is complicated (possibly due to its high fitness or harboring a selectable trait) [6]. To adapt to new culture conditions, the avirulent phenotype selectively accumulated [8,33]. The shape of the phylogenetic tree further reflected this phenomenon in which the *env*-Δ236D-phenotype gradually became predominant within the whole population but progressively lost its virulence. Correspondingly, the *env*-Δ236D-phenotype became dominant with ongoing passages, and thus the strain lost its virulence in animals.

In summary, our study characterized EIAV *env* quasispecies and observed a virulence-correlated parallel in *env* variation during EIAV passage in vitro. However, this strain in the context of the ensemble of the predominant population progressively lost its virulence in vivo. Hence, our data support the view of quasispecies theory and provide a role for the *env* quasispecies in EIAV pathogenesis, which partly explains the attenuation-related mechanism of EIAV.

## Figures and Tables

**Figure 1 viruses-11-00380-f001:**
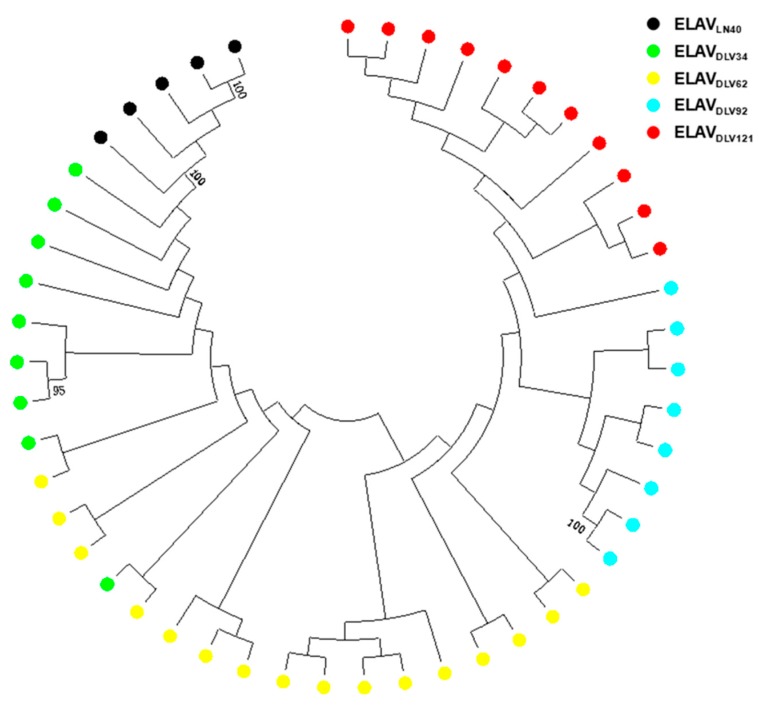
The evolutionary history was inferred by applying MEGA 7.0 program based on the general time reversible model. Whole genome sequences were obtained by bulk PCR. Bootstrap values above 80% are indicated. EIAV_LN40_, EIAV_DLV34_, EIAV_DLV62_, EIAV_DLV92_, and EIAV_DLV121_ are presented with black circles, black triangles, gray circles, blue circles and red circles, respectively.

**Figure 2 viruses-11-00380-f002:**
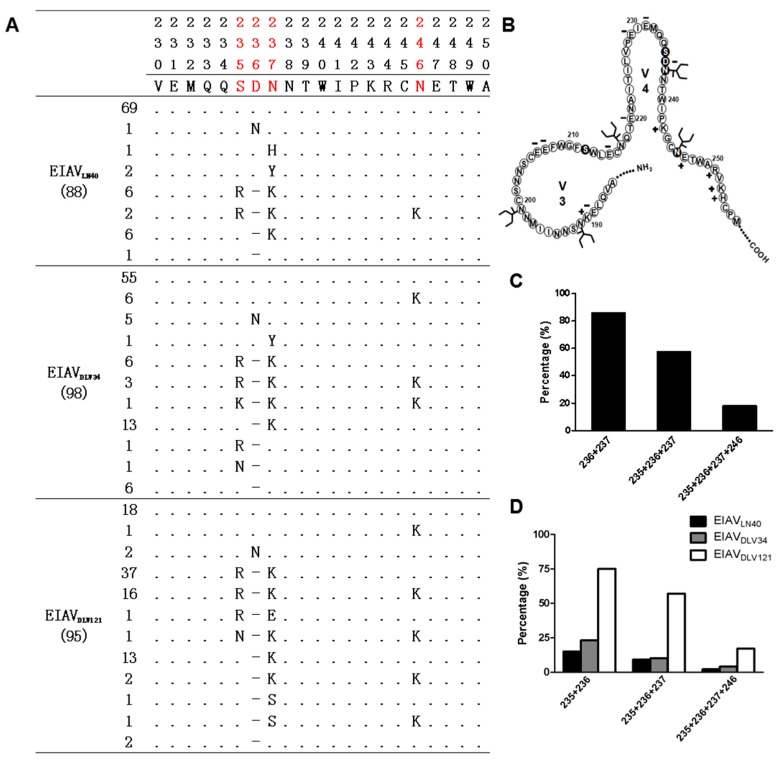
Distribution of the consensus mutations located in the V3 and V4 regions and schematic structural representations. (**A**) Red letters are defined as critical consensus mutation sites displayed within the Env region during the EIAV attenuation process. (**B**) Schematic figure of the EIAV_DLV121_ V3 and V4 regions. N-Glycosylation sites are shown as branched lines. (**C**) Comparison of the proportions of several combined mutations at 235, 236, 237, and 246 for the whole mutants. (**D**) The percentages of these combined mutations at the key stages of vaccine development (EIAV_LN40_, EIAV_DLV34_, and EIAV_DLV121_).

**Figure 3 viruses-11-00380-f003:**
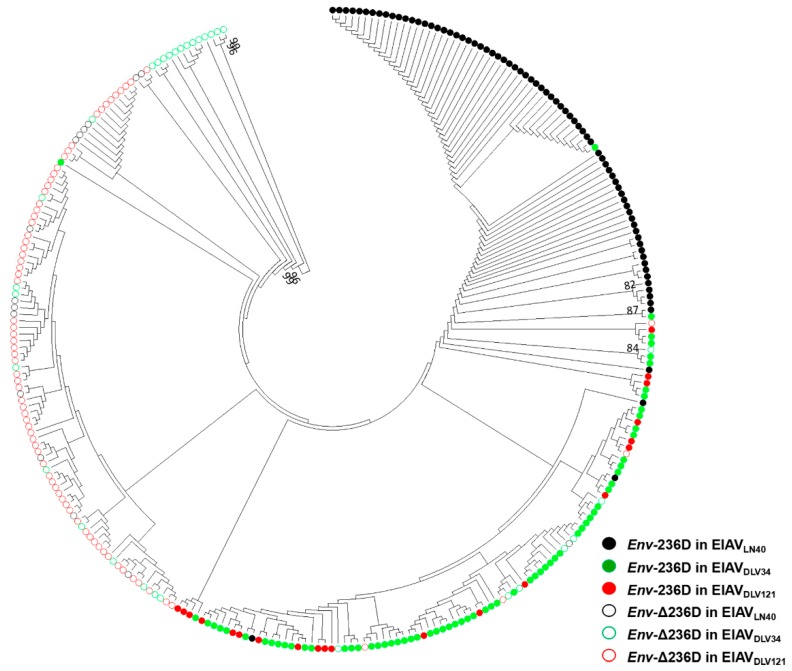
Construction of a maximum likelihood tree using the MEGA 7.0 program based on the Tamura-Nei model. Bootstrap values above 80% are indicated. EIAV_LN40_, EIAV_DLV34_, and EIAV_DLV121_ are shown in black, green and red, respectively. Solid and hollow circles represent the *env*-236D-phenotype and *env*-Δ236D-phenotype sequences, respectively. All samples were collected from three virus strains (EIAV_LN40_, EIAV_DLV34_, and EIAV_DLV121_). A total of 381 sequences were analyzed.

**Figure 4 viruses-11-00380-f004:**
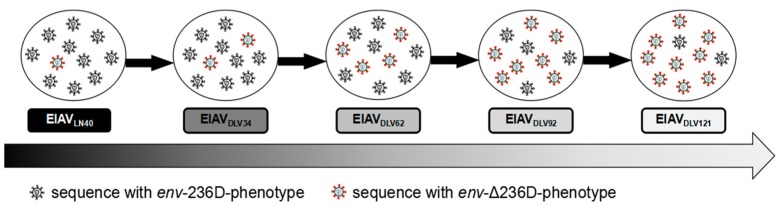
A schematic evolutionary model comparing the EIAV virulent strain to the vaccine strain during long-term passage in vitro. The virulent virus strains are presented with black circles, and the avirulent virus strains are presented with red circles. In this model, the *env*-Δ236D-phenotype sequences preexisted in EIAV_LN40_ as a minority population. With serial passages in vitro, sequences containing the *env*-Δ236D-phenotype gradually became the predominant EIAV quasispecies and generated high replicative fidelity under the new environmental selection conditions. Virulence was simultaneously attenuated as a result, as discussed in the text.

**Table 1 viruses-11-00380-t001:** Amino acid differences in Env V4 among different Equine Infectious Anima Virus (EIAV) strains obtained using bulk PCR.

	2	2	2	2	2	2	2	2	2	2	2	2	2	2	2	2	2	2	2	2	2	2	2	2	2	2	2	2	2	2	2	2	Mutation Rate (236D/-)	Morbidity (Horses)
	2	2	2	2	2	2	2	2	2	3	3	3	3	3	3	3	3	3	3	4	4	4	4	4	4	4	4	4	4	5	5	5		
	1	2	3	4	5	6	7	8	9	0	1	2	3	4	5	6	7	8	9	0	1	2	3	4	5	6	7	8	9	0	1	2		
EIAV_LN40_	N_29_	A_29_	I_29_	T_29_	I_29_	L_29_	V_29_	P_29_	E_29_	V_29_	E_29_	M_28_	Q_29_	Q_28_	S_29_	D_29_	N_28_	N_29_	T_29_	W_29_	I_29_	P_29_	K_29_	R_29_	C_27_	N_29_	E_29_	T_29_	W_29_	A_29_	R_29_	V_29_	0.00%	100%
												I_1_		R_1_			S_1_								Y_2_									
EIAV_DLV34_	S_9_	A_16_	I_15_	T_15_	I_16_	L_16_	V_16_	P_16_	E_16_	V_10_	E_16_	M_16_	Q_16_	Q_14_	S_16_	D_14_	N_16_	N_16_	T_16_	W_16_	I_16_	P_16_	K_16_	G_8_	C_16_	N_14_	E_13_	T_16_	W_16_	A_16_	R_12_	V_16_	6.25%	100%
	N_7_		V_1_	A_1_						I_6_				K_1_		N_1_								R_8_		K_2_	K_3_				K_4_			
														R_1_		-_1_																		
EIAV_DLV62_	N_9_	A_16_	I_16_	T_16_	I_16_	L_16_	V_16_	P_16_	E_16_	V_9_	E_16_	M_16_	Q_16_	Q_14_	S_16_	D_9_	N_16_	N_16_	T_16_	W_16_	I_16_	P_16_	K_16_	R_11_	C_16_	N_13_	E_13_	T_16_	W_16_	A_16_	R_12_	V_16_	18.5%	100%
	S_7_									I_7_				R_2_		N_4_								G_5_		K_3_	K_2_				K_4_			
																-_3_											G_1_							
EIAV_DLV92_	N_11_	A_19_	I_19_	T_16_	I_19_	L_19_	V_19_	P_19_	E_19_	V_18_	E_19_	M_19_	Q_19_	Q_17_	S_19_	D_3_	N_13_	N_18_	T_16_	W_19_	I_18_	P_19_	K_19_	R_17_	C_19_	N_14_	E_18_	T_18_	W_19_	A_19_	R_12_	V_19_	42.1%	9.1%
	S_8_			A_3_						I_1_				R_1_		N_8_	K_5_	D_1_	A_3_		V_1_			G_2_		K_5_	K_1_	A_1_			K_7_			
														K_1_		-_8_	R_1_																	
EIAV_DLV121_	N_11_	A_17_	I_18_	T_17_	I_18_	L_18_	V_18_	P_18_	E_17_	V_17_	E_17_	M_18_	Q_18_	Q_18_	R_15_	-_18_	K_17_	N_18_	T_18_	W_18_	I_17_	P_18_	K_18_	R_17_	C_18_	K_9_	E_10_	T_18_	W_18_	A_18_	K_13_	V_17_	100%	0
	S_7_	T_1_		A_1_					K_1_	I_1_	K_1_				S_3_		S_1_				V_1_			G_1_		N_8_	K_8_				R_5_	I_1_		
																										E_1_								

The additional amino acid D, which consistently appeared in virulent EIAV strains, is marked with dark grey, and N-linked glycosylation sites are marked with light grey.

**Table 2 viruses-11-00380-t002:** Amino acid differences in Env V4 among different EIAV strains detected using single genome amplification (SGA).

	2	2	2	2	2	2	2	2	2	2	2	2	2	2	2	2	2	2	2	2	2	2	2	2	2	2	2	2	2	2	2	2	Mutation Rate (236D/-)
	2	2	2	2	2	2	2	2	2	3	3	3	3	3	3	3	3	3	3	4	4	4	4	4	4	4	4	4	4	5	5	5	
	1	2	3	4	5	6	7	8	9	0	1	2	3	4	5	6	7	8	9	0	1	2	3	4	5	6	7	8	9	0	1	2	
EIAV_LN40_	N_86_	A_88_	I_88_	T_88_	I_88_	L_88_	V_88_	P_88_	E_85_	V_87_	E_88_	M_88_	Q_87_	Q_88_	S_80_	D_72_	N_71_	N_88_	T_88_	W_88_	I_88_	P_88_	K_87_	R_80_	C_88_	N_86_	E_87_	T_87_	W_88_	A_88_	R_81_	V_88_	17.0%
	S_2_								V_1_	I_1_			H_1_		R_8_	-_15_	K_14_						N_1_	G_8_		K_2_	K_1_	I_1_			K_7_		
									K_1_							N_1_	H_1_																
									Q_1_																								
EIAV_DLV34_	N_51_	A_97_	I_98_	T_98_	I_98_	L_97_	V_98_	P_97_	E_94_	V_60_	E_96_	M_98_	Q_98_	Q_82_	S_86_	D_63_	N_75_	N_98_	T_98_	W_95_	I_95_	P_97_	K_96_	R_57_	C_97_	N_85_	E_79_	T_96_	W_94_	A_94_	R_75_	V_94_	30.6%
	S_47_	T_1_				V_1_		L_1_	K_3_	I_38_	K_2_			K_8_	R_10_	-_30_	K_22_			S_1_	S_2_	L_1_	Q_1_	G_41_	W_1_	K_10_	K_17_	A_1_	C_3_	C_2_	K_20_	I_4_	
									D_1_					R_8_	K_1_	N_5_	Y_1_			C_2_	C_1_		P_1_			S_3_	G_2_	I_1_	R_1_	S_1_	T_3_		
															N_1_															T_1_			
EIAV_DLV121_	N_76_	A_95_	I_95_	T_92_	I_95_	L_95_	V_94_	P_95_	E_92_	I_10_	E_93_	M_94_	Q_94_	Q_91_	R_54_	-_74_	K_69_	N_95_	T_95_	W_95_	I_95_	P_95_	K_95_	R_68_	C_95_	N_74_	E_83_	T_94_	W_95_	A_93_	K_55_	V_95_	77.9%
	S_18_			A_3_			I_1_		K_1_	V_84_	K_1_	V1	H_1_	R_3_	S_40_	D_19_	N_23_							G_25_		K_21_	K_11_	A_1_		T_2_	R_40_		
	K_1_								D_1_	L_1_	V_1_			K_1_	N_1_	N_2_	S_2_							K_2_			G_1_						
									Q_1_								E_1_																
pLGFD3-8	N_100_	A_100_	I_100_	T_100_	I_100_	L_100_	V_100_	P_100_	E_99_	V_97_	E_99_	M_100_	Q_98_	Q_100_	S_87_	-_100_	K_100_	N_100_	T_97_	W_98_	I_99_	P_100_	K_100_	G_88_	C_99_	K_99_	K_99_	T_100_	W_99_	A_100_	R_97_	V_100_	100%
									Q_1_	I_3_	K_1_		H_1_		R_13_				I_3_	C_2_	V_1_			R_12_	G_1_	N_1_	E_1_		C_1_		K_2_		
													R_1_																		M_1_		

The additional amino acid D, which consistently appeared in virulent EIAV strains, is marked with dark grey, and N-linked glycosylation sites are marked with light grey.

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
