# Peer review of "Characterization of EIAV env Quasispecies during Long-Term Passage In Vitro: Gradual Loss of Pathogenicity"

_viruses, 2019, doi:10.3390/v11040380_

Round 1

Reviewer 1 Report

The manuscript “Characterization of EIAV gp90 quasispecies during long-term passage in vitro: gradual loss of pathogenicity” aims to describe quasispecies evolution of EIAV strains during a passage-based attenuation process.

While the manuscript is interesting, it is not always very easy to follow. In my opinion, the manuscript would need to be revised extensively prior to be considered for publication.

I would suggest the following:

- The authors would need to be more accurate in their objectives.

- Overall, some of the sentences would need to be rephrased and the English would need to be checked.

- Two different methods have been used; maybe the authors should focus on the most sensitive (and present the other one as supplementary data).

- Some of the figures and tables present roughly the same type of results (e.g. Table 1, Figure 2), therefore supplementary data should be used as frequently as possible to avoid redundancy.

- Table 1 reports morbidity results. Details of the animal work is not presented in this manuscript. References linked to this animal work is limited, incomplete or difficult to access. Taking into account that increase attenuation is a very important element of the study, all animal works and results should be easily accessible or well presented here.

Author Response

Response to Reviewer 1 Comments 

Point 1: The authors would need to be more accurate in their objectives.

Response: Thanks for the suggestions. We have modified Introduction, Result, and Discussion section in the revised manuscript to better demonstrate our finding. In brief, this paper systematically analyzed the dynamic changes of EIAV env during the passages in vitro, which fully reflects the quasispecies characteristics of EIAV. The main innovations in this study are: i) we observed a virulence-correlated parallel in env variation during the passages in vitro. ii) we certified one key mutation that appeared in the EIAV attenuated vaccine strains (termed vaccine-like env), which actually existed in EIAV virulent strains ; iii) the vaccine-like env gradually increased in the population during the in vitro passages; Therefore, we propose a hypothesis that the EIAV attenuated vaccine strain may be a small branch of the EIAV quasispecies.

Point 2: Overall, some of the sentences would need to be rephrased and the English would need to be checked.

Response: We apologize for it. We have sent this revised manuscript to the professional English language editing company (AJE) to improve the quality of this manuscript.

Point 3: Two different methods have been used; maybe the authors should focus on the most sensitive (and present the other one as supplementary data).

Response: We thank the reviewer’s great suggestion. In this study, we used two methods to elucidate the potential correlation between viral quasispecies evolution and pathogenesis. According to our results, SGA method has more sensitivity than double-probe real-time PCR. As suggested , we present this part as Figure S3.

Point 4: Some of the figures and tables present roughly the same type of results (e.g. Table 1, Figure 2), therefore supplementary data should be used as frequently as possible to avoid redundancy.

Response: We thank the reviewer’s suggestion. As suggested, we have changed the Figure 2 as Figure S2 to avoid redundancy in the revised manuscript. Considering the result presented in Table 1 was the premise of the design probe and also could provide a comparative analysis with SGA result. So we think it is better to present it as main result.

Point 5: Table 1 reports morbidity results. Details of the animal work is not presented in this manuscript. References linked to this animal work is limited, incomplete or difficult to access. Taking into account that increase attenuation is a very important element of the study, all animal works and results should be easily accessible or well presented here.

Response: Thanks for the suggestion and we make an apology for it. Because this result is derived from the historical data during this attenuated vaccine development. As suggested, we presented the results as Figure S1. And we have added the description about animal work in the figure legend.

Reviewer 2 Report

23-24: please check the English

53 and 55: is “shown” correct? Did you mean “showed”?

54: I suggest to do not use “And” at the benginning of the sense

51-61: I suggest to revise the English style to clarify the text. For non expert readers, I suggest to report that gp90 protein is codified by env gene (see lines 134-137).

81: please report the company producing the reagents used

96: please report the companyproducing thepMD18-T reagents

98: how many clones have been tested by PCR and subsequently sequenced?

112: here the name pLGFD3-8 is reported for the probe; however, at line 101 only probe 3-8 is reported: please, clarify the name of the probe.

126: please report the GenBank accession numbers

129: model tests and bootstrap values are not reported

132: Authors instructions of Viruses journal (https://www.mdpi.com/journal/viruses/instructions) report that separate sections should be provided for results and for discussion; conclusion section is not mandatory, while here a “Results and Discussion” section is reported. On the basis of Editor's opinion, the style of these sections could be maintained or changed following the journal instructions.

134-137: these parts fit better in the introduction section

134: you already have used “env” and “(env)” is not required here but, in case, the first time that you use “env”.

135: please check if “while” is appropriate here

142-143: please check the English to clarify the sentence

143: “was consistence” did you mean “was consistant”?

164: “shown” is not correct

193: please check the English

229: as presented “in”

230: please check the English for the verb “formed”

238: please report the reference for LTR investigation. No space before dot

252: “in” Figure 6

265-266: please check, the sentence is not clear

271: comparison of

295: I did not find instructions for the reference list reporting that “et al.” should be used to replace the list of Authors. Please, check the reference style for this journal.

Author Response

Response to Reviewer 2 Comments

Point 1: 23-24: please check the English

Response: Thanks for your suggestion. We have change “Among…… the 236-deleted in V4 region of gp90 resulted….. site which was crucial for virulent” to “Among these mutations, deletion of amino acid 236 in the V4 region of env resulted in the loss of one N-glycosylation site that was crucial for virulence” (line 26-28).

Point 2: 53 and 55: is “shown” correct? Did you mean “showed”?

Response: We have corrected the errors using “showed” instead of ‘shown’ (line 57).

Point 3: 54: I suggest to do not use “And” at the beginning of the sense

Response: Thanks for your suggestion. We have deleted the “And” in the revised manuscript (line 58).

Point 4: .51-61: I suggest to revise the English style to clarify the text. For non expert readers, I suggest to report that gp90 protein is codified by env gene (see lines 134-137).

Response: Thank you and we have use the term “env” throughout this article to avoid the confusing.

Point 5: 81: please report the company producing the reagents used

Response: We have added the company name (line 87).

Point 6: 96: please report the company producing the pMD18-T reagents

Response: We have added the company name (line 103).

Point 7:  98: how many clones have been tested by PCR and subsequently sequenced?

Response: We have added the numbers of clones sequenced (line 105).

Point 8: 112: here the name pLGFD3-8 is reported for the probe; however, at line 101 only probe 3-8 is reported: please, clarify the name of the probe.

Response: We have use FAM-probe instead of probe 3-8 which targeted the env-Δ236D-phenotype sequence and HEX-probe instead of probe LN40 targeting the env-236D-phenotype sequence in revised manuscript (line 119-120).

Point 9: 126: please report the GenBank accession numbers

Response: We have added the GenBank accession numbers in revised manuscript (Line 135-136).

Point 10: 129: model tests and bootstrap values are not reported

Response: Thanks and according the review’s suggestion, we have construct a maximum likelihood tree using the MEGA 7.0 program, add the model tests and bootstrap values in ML tree and update the Figure 1 and Figure 3.

Point 11: 132: Authors instructions of Viruses journal (https://www.mdpi.com/journal/viruses/instructions) report that separate sections should be provided for results and for discussion; conclusion section is not mandatory, while here a “Results and Discussion” section is reported. On the basis of Editor's opinion, the style of these sections could be maintained or changed following the journal instructions.

Response: Thanks for your comments. We have modified the Results and Discussion sections in the revised manuscript according to the journal instructions.

Point 12: 134-137: these parts fit better in the introduction section

Response: Thanks for your suggestion. This sentence has been removed in the revised manuscript.

Point 13: 134: you already have used “env” and “(env)” is not required here but, in case, the first time that you use “env”.

Response: We apologize for the confusing. As the comment-12 suggestion, this sentence has been removed.

Point 14: 135: please check if “while” is appropriate here

Response: This sentence has been removed has been deleted in the revised manuscript.

Point 15: 142-143: please check the English to clarify the sentence

Response: We have change the sentence “Obviously, individual clusters formed that were accompanied by parallel changes in virulence” to “Obviously, individual clusters accompanied by a decrease in virulence were generated” in the revised manuscript (line 148 to 149).

Point 16:143: “was consistence” did you mean “was consistant”?

Response: Thank you. We have corrected the term to “consistant” (line 149).

Point 17: 164: “shown” is not correct

Response: Thank you. We have changed ' shown ' to ' shows' in revised manuscript.

Point 18: 193: please check the English

Response: We have modified this sentence to “Thus we were interested in using SGA to resequence env to obtain more information about env quasispecies during long-term passage” (line 189 to 190). 

Point 19:229: as presented “in”

Response: We have corrected this error.

Point 20: 230: please check the English for the verb “formed”

Response: We have change this term to “evolved (line 228).

Point 21: 238: please report the reference for LTR investigation. No space before dot

Response: Thanks and we have added the reference here and also deleted the space (line 237-238).

Point 22: 252: “in” Figure 6

Response: We apologize for the error and we have corrected it.

Point 23: 265-266: please check, the sentence is not clear

Response: Thanks and we have deleted this sentence.

Point 24: 271: comparison of

Response: We have corrected this error.

Point 25: 295: I did not find instructions for the reference list reporting that “et al.” should be used to replace the list of Authors. Please, check the reference style for this journal.

Response: We have clarified the reference style in the revised version of manuscript according to the requirements of the journal.

Reviewer 3 Report

In this paper, Liu and colleagues perform sequencing and evolutionary analyses on an EIAV quasispecies population that was passaged 121 times, resulting in an avirulent vaccine strain. They demonstrate that decreased virulence was associated with increased abundance of an amino acid deletion and two amino acid substitutions in the virus envelope protein. They also demonstrate using single-genome analyses that these mutations were pre-existing in the virulent virus prior to passage and that they sequentially increased over time with passage. The laboratory methods used in this study appear to be appropriate. However, I believe the data analyses, particularly the phylogeny, need to be improved in order to justify conclusions regarding the evolution of this virus population. I also find the manuscript to be very difficult to understand in many places, which makes it difficult to interpret. 

1. Spelling and grammar throughout the manuscript should be improved to enhance interpretability and impact.

2. The analyses demonstrate that three mutations are selected in culture over time. However, there is no analysis of whether these mutations are linked, e.g. occurring in the same genomes. An analysis of linkage might be informative in terms of function.

3. The most concerning aspect of the Fig 5 phylogenetic analysis is that there is no method used to test how reliable the branches are on the tree. This is typically performed for neighbor-joining and maximum likelihood trees by bootstrapping. Without this analysis, there is no way of knowing whether the groupings shown on the trees are good or bad estimates.

4. Maximum likelihood is generally considered a better method for constructing these type of trees than Neighbor joining. I would suggest performing a maximum likelihood tree with a program such as PhyML.

5. The nucleotide substitution models chosen (Kimura 2-parameter) may not appropriate for lentiviruses. I would suggest estimating the substitution model directly from your data using free tools such as those available through www.datamonkey.org.

6. Given the high propensity of lentiviruses for recombination, you should test your sequences for evidence of recombination prior to performing phylogeny.

7. The sequential trees shown in Fig 5B are not useful – preparing three separate trees for three timepoints (passages) does not provide any information on the evolutionary relationship between the sequences from different times (see next point below).

8. Since you know the actual “time” of each of these samples (passage number), you could construct a time-calibrated phylogeny using Bayesian phylogenetics implemented in a program such as BEAST using passage number as time. This would give you a more realistic reconstruction of the evolution of this population. For instance, clearly the 236 deletion is being selected for in the population. But are all the sequences containing this mutation derived from the clones present in the original LN40 virus or did new 236 deletions occur and were also selected for? When did these new mutations occur? A time structured tree would help estimate this and be a great replacement for the trees provided in Figure 5B.

9. Line 27 in the abstract, “Natural quasispecies” is unclear, please restate. Do you mean to say that the mutations selected for by passage are also found naturally in EIAV-infected horses?

10. Line 52, unclear what is meant by “high variations”

11. Line 56, “because the LTR is a noncoding region, the data are piecemeal.” Unclear what is meant by this statement.

12. Line 231 refers to black and red triangles on Fig 5a. There are no triangles on Fig 5a.

13. Line 241-244, “The env-236D phenotype is mainly distributed on the right half of the phylogenetic tree, whereas the env-delta236D phenotype is distributed on the opposite side. The two typical env phenotypes shown here obviously alternated conversion between adjacent or nonadjacent generations.” I do not understand what these statements are attempting to convey – please clarify.

Author Response

Response to Reviewer 3 Comments

Point 1: Spelling and grammar throughout the manuscript should be improved to enhance interpretability and impact.

Response: We apologize for it. We have sent this revised manuscript to the professional English language editing company (AJE) to improve the quality of this manuscript.

Point 2: The analyses demonstrate that three mutations are selected in culture over time. However, there is no analysis of whether these mutations are linked, e.g. occurring in the same genomes. An analysis of linkage might be informative in terms of function.

Response: We thank the reviewer’s great suggestion. We have added the results of the mutations linkage analysis through calculating the frequency of key mutations235236237 and 246in combination during the attenuated progress. As expected, we observed the percentage of the combined mutations at 236 and 237 is about 85%. We speculated that there existed a linkage between the site 236 and 237. This result further supports our conclusion. Also as the reviewer’s suggestion, we have added the results as Figure 2C and 2D in the manuscript and updated the correlated parts in results and discussion sections.

Point 3: The most concerning aspect of the Fig 5 phylogenetic analysis is that there is no method used to test how reliable the branches are on the tree. This is typically performed for neighbor-joining and maximum likelihood trees by bootstrapping. Without this analysis, there is no way of knowing whether the groupings shown on the trees are good or bad estimates.

Response: We thank the reviewer’s great suggestion. We have performed phylogenetic analysis for maximum likelihood trees by bootstrapping in the revised manuscript.

Point 4: Maximum likelihood is generally considered a better method for constructing these type of trees than Neighbor joining. I would suggest performing a maximum likelihood tree with a program such as PhyML.

Response: We thank the reviewer’s great suggestion. In the revised version of manuscript, we have constructed a maximum likelihood tree using the MEGA 7.0 software and updated Figure.

Point 5: The nucleotide substitution models chosen (Kimura 2-parameter) may not appropriate for lentiviruses. I would suggest estimating the substitution model directly from your data using free tools such as those available through www.datamonkey.org.

Response: We thank the reviewer’s suggestion. Our data was firstly estimated to choose the best model by Find Best DNA/Protein models (ML) program of MEGA 7.0 and then constructed the phylogenetic tree. For bulk PCR data, we used the General Time Reversible Model and Tamura-Nei model was chosen for SGA data analysis.

Point 6: Given the high propensity of lentiviruses for recombination, you should test your sequences for evidence of recombination prior to performing phylogeny.

Response: We thank the reviewer’s great suggestion. We have detected and analyzed the recombination patterns in all sequences used in this study performed by RDP4 v4.97 (http://web.cbio.uct.ac.za/~darren/rdp.html). The result showed that 4 of 284 clones existed recombination event (these four clones all derives from the EIAVDLV34). However, the recombination regions were mainly distributed from site 710 to 888 aa, which are not involved in the hyper variable regions we concerned in this study. Hence we proposed that the recombination would not affect the results and conclusion.

Point 7: The sequential trees shown in Fig 5B are not useful preparing three separate trees for three timepoints (passages) does not provide any information on the evolutionary relationship between the sequences from different times (see next point below).

Response: We agree with the reviewer’s opinions. We have deleted this part (three separate trees for three time-points), update the Figure 3 with the maximum likelihood tree using the MEGA 7.0 software and add the corresponding figure legend in the revised manuscript.

Point 8: Since you know the actual “time” of each of these samples (passage number), you could construct a time-calibrated phylogeny using Bayesian phylogenetics implemented in a program such as BEAST using passage number as time. This would give you a more realistic reconstruction of the evolution of this population. For instance, clearly the 236 deletion is being selected for in the population. But are all the sequences containing this mutation derived from the clones present in the original LN40 virus or did new 236 deletions occur and were also selected for? When did these new mutations occur? A time structured tree would help estimate this and be a great replacement for the trees provided in Figure 5B.

Response: We thank the reviewer’s suggestion. And we also try to construct a time-calibrated phylogeny using Bayesian phylogenetics. But in fact it is very difficult to perform it because we have only three time points in this study. So we feel very sorry not to realize it. While according to the result as described in Figure 3, the env-Δ236D-phenotype in EIAVLN40 distributed on the same branch with EIAV vaccine strain. Hence we could speculate that the sequences containing the 236 deletion generated from the original EIAVLN40 strain and accumulated during the long passage. In the future experiment, we are interested in choosing more timepoints to construct a time-calibrated phylogeny to further investigate the problem.

Point 9: Line 27 in the abstract, “Natural quasispecies” is unclear, please restate. Do you mean to say that the mutations selected for by passage are also found naturally in EIAV-infected horses?

Response: We apologize for this confusing. We mean EIAV attenuated vaccine may be selected from the EIAV quasispecies during adaptation to altered microenvironments. So in the revised version manuscript, we have change “our results suggested the attenuated vaccine may be a natural quasispecies of EIAVLN40” to “EIAV attenuated vaccine may originate from a branch of quasispecies of EIAVLN40” (line 30 to 31).

Point 10: Line 52, unclear what is meant by “high variations”

Response: We used the term “high variations” to emphasize 8 regions have high mutation rate (Xue-Feng Wang, et al., Virus, 2011). The terms have been changed to “hyper variable regions” in the revised version (line 56 to 57).

Point 11: Line 56, “because the LTR is a noncoding region, the data are piecemeal.” Unclear what is meant by this statement.

Response: We apologize for the confusing. In the revised manuscript, we have delete the sentence “However, LTR is a noncoding region which could not reflect predominant antigen gene of EIAV evolutionary pattern.” and restate the sentence.

Point 12: Line 231 refers to black and red triangles on Fig 5a. There are no triangles on Fig 5a.

Response: We apologize for the error and we have changed “triangles” to “solid and hollow circles” in the Figure 3 (line 239).

Point 13:. Line 241-244, “The env-236D phenotype is mainly distributed on the right half of the phylogenetic tree, whereas the env-delta236D phenotype is distributed on the opposite side. The two typical env phenotypes shown here obviously alternated conversion between adjacent or nonadjacent generations.” I do not understand what these statements are attempting to convey – please clarify.

Response: We apologize for the confusing. In the revised version, we have restated this part. “Over the course of 121 generations in vitro, the cluster distribution of related sequences taken from EIAVDLV121 showed a pattern similar to that of EIAVLN40 except for reversal of the predominant phenotype from env-236D to env-Δ236D at the individual population level. We observed two typical env phenotypes displayed a gradual transition pattern”. This phenomenon suggests that evolution of the EIAV attenuated vaccine in vitro involves the gradual accumulation of dominant quasispecies (line 232 to 235).

Reviewer 4 Report

Reviewer comments:

This manuscript reports characterization of EIAV gp90 strains showed gradual loss of their pathogenicity during long-term passages in vitro. By sequencing analysis, several key mutations occurred within gp90 accompanied with increased passage and attenuated EIAV variants. One critical amino acid in V4 region of gp90 resulting in the loss of N-glycosylation site was observed in wild EIAVLN40 strain and passage EIAV strains also. This mutation (Δ236D) is important for the viral virulence which is increased percentage during serial passages. The authors also used several methods to confirm the mutation in different strains. The interesting point is that EIAV vaccine strains may preexist in the EIAV population as a minority population and the virulence of EIAV strain could be reduced if it is passaged serially in vitro.

Major

Was the attenuated EIAV strain probably selected within the mixed population but not evolved in cultured cells during long-term passages?  Does an infection with the pathogenic virus using molecular clone induce attenuated viruses in vitro?

Did two phenotype viruses between env-236D and env-Δ236D show different viral properties about viral proliferation and CPE induction in vitro? Did these viral properties associate with the outcome?

Minor

In the material and methods

Study subject:  The authors should mention how they passage the virus to show the attenuated strains. Because this reviewer can’t read ref.17, please explain this sentence on line 71-72.

Data processing and analysis: it needs references.

line 231:triangle should be circle in the figure 5A.

Author Response

Response to Reviewer 4 Comments

Point 1: Was the attenuated EIAV strain probably selected within the mixed population but not evolved in cultured cells during long-term passages?  Does an infection with the pathogenic virus using molecular clone induce attenuated viruses in vitro?

Response: Our results strongly suggest that the attenuated EIAV strain probably exited within the mixed population, but under selection and evolution during long-term in vitro passages in cultured cells. Unfortunately, we have no pathogenic molecular clone used to test if it can evolve to attenuated virus in vitro. In the history, it took about 4 years to finish the attenuation procedure of EIAV in vitro, and each generation of the virus need to be carefully evaluated and selected. Thus, we are not sure if we can get attenuated virus by passage a pathogenic molecular EIAV clone in vitro.  

Point 2: Did two phenotype viruses between env-236D and env-Δ236D show different viral properties about viral proliferation and CPE induction in vitro? Did these viral properties associate with the outcome?

Response: In fact, our previously study showed that the molecular clones with env-236D or env-Δ236D had the similar replication kinetics patterns in eMDMs (equine monocyte-derived macrophage) (Xiue Han, Virus Genes 2016). However, we also noticed that the virus with env-236D showed the enhance resistance to serum neutralizing antibodies compared with the virus with env-Δ236D (Xiue Han, Virus Genes 2016).

Point 3: Study subject:  The authors should mention how they passage the virus to show the attenuated strains. Because this reviewer can’t read ref.17, please explain this sentence on line 71-72.

Response: We thank the reviewer’s great suggestion. We have added the relative content in the revised manuscript and update this information with one reference (line 75 to 78).

Point 4: Data processing and analysis: it needs references.

Response: We have added the references in the revised manuscript (line 140).

Point 5: line 231:triangle should be circle in the figure 5A.

Response: We apologize for the error and we have change “triangles” to “solid and hollow circles” in Figure 3 (line 239).

Round 2

Reviewer 1 Report

thank you for your answers.

Reviewer 3 Report

The authors made strong efforts to incorporate changes based on reviewers' comments and have corrected the major issues with the manuscript. I recommend acceptance in present form.

Reviewer 4 Report

I accept the author's response.

Viruses EISSN 1999-4915 Published by MDPI AG, Basel, Switzerland RSS E-Mail Table of Contents Alert
Back to Top